# Information-Theoretic Approach to Detect Collusion in Multi-Agent Games

**Trevor Bonjour**[1]                **Vaneet Aggarwal**[2]                **Bharat Bhargava**[1]

[1]Department of Computer Science, Purdue University, West Lafayette, Indiana, USA
[2]School of Industrial Engineering, Purdue University, West Lafayette, Indiana, USA

## Abstract

Collusion in a competitive multi-agent game occurs when two or more agents co-operate covertly to the disadvantage of others. Most competitive multi-agent games do not allow players to share information and explicitly prohibit collusion. In this paper, we present a novel way of detecting collusion using a domain-independent information-theoretic approach. Specifically, we show that the use of mutual information between actions of the agents provides a good indication of collusive behavior. Our experiments show that our method can detect varying levels of collusion in repeated simultaneous games like iterated Rock Paper Scissors. We further extend the detection to partially observable sequential games like poker and show the effectiveness of our methodology.

## 1 INTRODUCTION

Recently, there has been growing interest in developing mechanisms to train agents to co-operate in multi-agent games [Tampuu et al., 2017, Jaques et al., 2019, Celli et al., 2019]. Many multi-agent games, however, do not allow co-operation between players. Collusion in a competitive multi-agent game occurs when two or more agents co-operate covertly, often to the detriment of others. Collusion poses a major threat [Yampolskiy, 2008, Yan, 2003, Yan and Randell, 2005] in competitive multi-agent games since the general assumption is that the players play to maximize their utility and it is often impossible to prevent some forms of collusion [Smed et al., 2006], especially in online settings. Detecting collusion in real-time is a difficult task as it often requires discerning and understanding a player's motivation. For this reason, collusion detection usually happens *post-hoc*. Yan [2010] motivates the need to design automatic solutions to detect collusion from historical game records.

In this work, we propose a novel information-theoretic approach to detect collusion amongst players given a sequence of records over several games. We use the game records to inform us about the strategies of different players across multiple games. We hypothesize that when two players collude, the effect they have on each other's strategies would be larger than if they were not colluding. This forms the basis of our collusion detection method. We formally describe our approach in Section 3. To evaluate our method, we conduct proof of concept experiments for perfect information simultaneous games and imperfect information sequential games. To test collusion detection for these games, we develop environments for a three-player iterated Rock Paper Scissors and three-player Leduc Hold'em poker. We show that our method can successfully detect varying levels of collusion in both games. We also report accuracy and swiftness [Smed et al., 2007] of our detection algorithm for different scenarios. For this paper, we limit the scope of our experiments to settings with exactly two colluding agents.

Collusion can take many forms. Smed et al. [2006] provide a classification of different forms of collusion that can take place among players. For multi-agent games, they define three types of collusion: spectator collusion, assistant collusion, and association collusion. In spectator collusion, a spectating player provides important information to their colluding partner. For example, in the first-person shooter game Counter-Strike, a dead player can move around in ghost mode and provide information about the location of other players. Assistant collusion is one in which the colluding partner doesn't aim to win but assists another player gain an advantage. For example, in Monopoly, a player could buy properties that prevent the non-colluding player from gaining a complete set of colored properties. A complete set of colored properties is required to collect rent from other players. Association collusion is where colluding players are in a symbiotic relationship such that each partner plays to benefit the other. For instance, in poker, colluding partners play more aggressively when either of them has a good hand, and play cautiously otherwise. We focus on detect-

*Accepted for the 38th Conference on Uncertainty in Artificial Intelligence* (UAI 2022).

ing collusion between active players in the game since it is improbable that data on spectators of the game is collected.

To the best of our knowledge, there is no publicly available data set for multi-agent games with known collusion among players. For this reason, we design hand-crafted collusion strategies for our experiments. We develop strategies for assistant collusion in iterated Rock Paper Scissors and association collusion in Leduc Hold'em poker. We show that our proposed method can detect both assistant and association collusion. Apart from rule-based collusion, we use Deep Reinforcement Learning [Arulkumaran et al., 2017] techniques to automatically construct different collusive strategies for both environments. Our method can successfully detect cooperative collusion in all scenarios where the agents use mixed strategies. We summarize our contributions as follows:

- We propose a novel information-theoretic approach to detect collusion (Section 3) in multi-agent games from historical game records.

- We develop rule-based colluding agents for multi-agent environments: three-player iterated Rock Paper Scissors (Section 5.1) and three-player Leduc Hold'em poker (Section 5.2). We generate automatic collusion between agents using Deep Reinforcement Learning.

- Our experiments show that the collusion detection method can detect different forms and multiple levels of collusion for fully observable simultaneous and partially observable sequential games (Section 5).

## 2 BACKGROUND

### 2.1 MULTI-AGENT MARKOV GAMES

In this work, we consider a multi-agent extension of Markov decision processes (MDPs) called Markov games [Littman, 1994]. We consider the scenario where multiple games are played between $n$ agents, for example, in a tournament setting. We restrict ourselves to scenarios where we have two colluding partners playing against other agents. Thus, we only consider settings where $n > 2$. The state of the environment is given by $s \in S$, where $S$ is the set of all possible states. At each timestep $t$, an agent $i$, with current state $s_t^i$ selects an action $a_t^i \in A$, where $A$ is the set of all possible actions. For a given agent, $A_i$ denotes the set of all actions taken by the agent. For a pair of agents $i,j$ we denote their joint state as $s_t^{ij}$ which is the state of both agents i and j taken together. The actions of all $n$ agents are combined to form a joint action $\boldsymbol{a}_t = [a_t^0, \ldots a_t^n]$. Each agent receives a reward $r_t^i(\boldsymbol{a}_t, s_t)$. A history of these values over time is termed as a trajectory, $\tau = \{s_t, \boldsymbol{a}_t, r_t\}$. Solving an MDP yields a policy $\pi$, which is a mapping from states to actions.

### 2.2 MUTUAL INFORMATION

Mutual information $\mathcal{I}(X;Y)$ of two discrete random variables $(X, Y) \sim p(x, y)$ is defined as:

$$\mathcal{I}(X;Y) = \sum_{\substack{x \in X \\ y \in Y}} p(x, y) \log \frac{p(x, y)}{p(x)p(y)} \qquad (1)$$

For discrete random variables $(X, Y, Z) \sim p(x, y, z)$, the conditional mutual information $\mathcal{I}(X;Y|Z)$ between $X$ and $Y$ given $Z$ is defined as:

$$\mathcal{I}(X;Y|Z) = \sum_{z \in Z} p(z)\mathcal{I}(X;Y|Z = z) \qquad (2)$$

## 3 APPROACH

Ideally, if all agents compete, their policies for a given state of the environment would be independent of each other. In practical scenarios, however, the policy of one agent might influence the policy of another agent. For instance, a competitor could inform their actions based on the actions of other players. The difference between a competitor and a colluding player is that the competitor would not change their actions based on the opponent i.e. for a given state and a given set of actions by the other players, they would play the same way. However, a colluding player, on the other hand, would play differently in the same perceived state against a non-colluding agent than they would against the colluding partner. In other words, the case where two agents collude, their individual influence on each other's policy would be larger than that of a non-colluding agent. We use this intuition as the basis of our collusion detection method. We define *individual influence* ($\gamma$), as the influence that one agent's policy has on the other. We define *net influence* ($\Gamma$) as the difference in the individual influence of one agent and the maximum individual influence other agents have on the said agent.

As seen in Section 2.1, the states and actions are associated with timestep $t$. To simplify the notation, we omit using the subscript $t$ going forward. Suppose we have $n$ agents and a set of their state-action pairs for a fixed number of game episodes. From the state-action pairs we can construct the empirical policy matrix for the agents. $\pi$ is a $S \times A$ matrix where each element gives the probability of taking an action $a \in A$ in a given state $s \in S$. $\pi^i$ represents the policy matrix for agent $i$ and $\pi^j$ represents the policy matrix for agent $j$. Each element of $\pi^i$ and $\pi^j$ give $p(a^i|s^i)$ and $p(a^j|s^j)$. We define $s^{ij}$ as the joint state of agent $i$ and agent $j$. The joint policy $\pi^{ij}$ gives a mapping from the joint state to probability of taking actions $a^i, a^j \in A$ for a given joint state $s^{ij}$. Each element of $\pi^{ij}$ gives $p(a^i, a^j|s^{ij})$. To measure the individual influence of $i$ on $j$, we need to capture how different the joint policy $\pi^{ij}$ is from $\pi^i \pi^j$. In other words, individual

influence captures how different the joint policies of the two agents are from what it would be if they were independent. If the two policies were conditionally independent, we would have $p(a^i, a^j|s^{ij}) = p(a^i|s^{ij})p(a^j|s^{ij})$. In the general setting (which includes partially observable games), the assumption is that the agents don't have access to the internal state of the other agent and hence don't have access to complete joint state $s^{ij}$. Hence, their actions should be independent of the internal state of the other agent and $p(a^i|s^{ij})$ simplifies to $p(a^i|s^i)$. For sequential games, we implicitly take the actions of the other players that have occurred previously, as part of the state, thus we omit explicitly including it in the probability equation. Thus, we have $p(a^i, a^j|s^{ij}) = p(a^i|s^i)p(a^j|s^j)$. To capture this difference between the distributions, we use the concept of conditional mutual information. We have:

$$\gamma(i;j) = \sum_{s^{ij} \in S^{ij}} p(s^{ij}) \sum_{\substack{a^i \in A^i \\ a^j \in A^j}} p(a^i, a^j|s^{ij}) \times$$

$$\log \frac{p(a^i, a^j|s^{ij})}{p(a^i|s^i)p(a^j|s^j)} \quad (3)$$

The net influence, $\Gamma(i;j)$, of $i \in n$ on $j \in n$ is defined as:

$$\Gamma(i;j) = \gamma(i;j) - \max_{\substack{k \in n \\ k \neq i,j}} \gamma(k;j) \quad (4)$$

To get the second term on the right-hand side of Equation (4), we calculate the individual influence on $j$ of all the other agents in $n$, that are not $i$ or $j$ and take the maximum value. For collusion to occur, both agents should have a positive net influence on each other. We say there is collusion between agents $i$ and $j$ if:

$$\Gamma(i;j) \text{ and } \Gamma(j;i) \geq \alpha \quad (5)$$

where $\alpha$ is the collusion threshold.

### 3.1 FULLY OBSERVABLE SIMULTANEOUS GAMES

In a fully observable environment, the observed state is common to all agents: $s^{ij} = s^i = s^j = s$. Equation (3) thus becomes:

$$\gamma(i;j) = \sum_{s \in S} p(s) \sum_{\substack{a^i \in A^i \\ a^j \in A^j}} p(a^i, a^j|s) \log \frac{p(a^i, a^j|s)}{p(a^i|s)p(a^j|s)}$$

$$\quad (6)$$

$$= \sum_{s \in S} p(s) \sum_{a^i \in A^i} p(a^i|s) \sum_{a^j \in A^j} p(a^j|a^i, s)$$

$$\times \log \frac{p(a^j|a^i, s)}{p(a^j|s)} \quad (7)$$

Effectively, for a fixed state $s$, the individual influence provides a measure of the difference between the probability of agent $j$ selecting an action $a^j$ given the state $s$ and $a^i$ (agent $i$'s action), and the probability of agent $j$ selecting the action $a^j$ given the state $s$. In other words, it measures how the actions of $i$ affect the actions of $j$. For a simultaneous game, all agents take an action at the same time. Since there is no sequence in which the individual agents take an action, the individual influence is symmetric: $\gamma(i;j) = \gamma(j;i)$. Note that $\Gamma(i;j) \neq \Gamma(j;i)$. For net influence $\Gamma(i;j)$ we have

$$\Gamma(i;j) = \gamma(i;j) - \max_{\substack{k \in n \\ k \neq i,j}} \gamma(k;j) \quad (8)$$

$$= \gamma(j;i) - \max_{\substack{k \in n \\ k \neq i,j}} \gamma(k;j) \quad (9)$$

and for net influence $\Gamma(j;i)$ we have:

$$\Gamma(j;i) = \gamma(j;i) - \max_{\substack{k \in n \\ k \neq i,j}} \gamma(k;i) \quad (10)$$

$$= \gamma(i;j) - \max_{\substack{k \in n \\ k \neq i,j}} \gamma(k;i) \quad (11)$$

Note that even though the first term on the right-hand side in Equation (10) is the same as in Equation (9), the second terms are different. In the first case, the second term denotes the maximum individual influence of other agents on agent $j$, while in the second case it denotes the maximum individual influence of other agents on agent $i$.

### 3.2 PARTIALLY OBSERVABLE SEQUENTIAL GAMES

For partially observable games, each agent's observed state is different: $s^i \neq s^j$. Thus, to calculate the individual influence, we use Equation (3) directly. In sequential games, unlike simultaneous games, there is an order in which the agents take an action. Suppose we have two agents $i$ and $j$, where $i$'s turn happens before $j$'s turn. Now when $i$ makes a move, this information is available to $j$ before $j$ makes a move. Along with capturing the effect of one agent's action on the other, the individual influence in the partially observable sequential case also captures the effect of having access to the joint state as compared to only having access to the individual state. Note that since this is a sequential game, the individual influence will not be symmetric: $\gamma(i;j) \neq \gamma(j;i)$. This is because when calculating the individual influence we need to take into account the sequence in which the actions were taken. When calculating $\gamma(i;j)$, we need to consider only the cases where action $a^j$ is executed after action $a^i$ in a game. Similarly, for $\gamma(j;i)$ we need to consider only the cases where action $a^i$ is executed after action $a^j$ in a game.

# 4 COLLUSION DETECTION ALGORITHM

We state the collusion detection problem as: Given a sequence of $m$ game records for $n$ agents, determine if two players collude and return the colluding pair of players. Each game record consists of a sequence of tuples (state, actions, reward) from the sequence of actions performed by the agents during a game. We give the main steps of the procedure in Algorithm 1 in Appendix B.

The first step for our algorithm is to construct a policy matrix $\pi^i$ for each agent $i \in n$ from these records. To construct $\pi^i$, we need to estimate $p(a^i|s^i)$. We use a hash table to implement the Monte Carlo method (MC) for estimating the distributions accurately. For $p(a^i|s^i)$, we have:

$$p(a^i|s^i) \equiv \frac{N(a^i, s^i)}{N(s^i)} \tag{12}$$

where $(a^i, s^i)$ is the number of times the action-state pair $(a^i, s^i)$ occur in the data and $N(s^i)$ is the total number of times the agent visits state $s^i$.

The next step is to construct the joint policy matrix for every pair of agents $i, j \in n$. We use MC sampling to calculate $p(a^i, a^j|s^{ij})$. We have:

$$p(a^i, a^j|s^{ij}) \equiv \frac{N(a^i, a^j, s^i, s^j)}{N(s^i, s^j)} \tag{13}$$

where $N(a^i, a^j, s^i, s^j)$ is the number of times $(a^i, a^j, s^i, s^j)$ occur in the data. Note that for sequential games, the ordering of $a^i$ and $a^j$ matters. When calculating $N(a^i, a^j, s^i, s^j)$ and $N(s^i, s^j)$, we only consider cases where agent $j$ takes an action after agent $i$.

Once the policy matrices are constructed we calculate and store the pair-wise individual influence $\gamma(i; j)$ for $i, j \in n$ for every pair of agents using Equation (3). The next step is then to calculate the net influence $\Gamma(i; j)$ for $i, j \in n$ for every pair of agents using Equation (4). If for exactly one pair of agents the net influence on each other exceeds the collusion threshold $\alpha$ (Equation (5)), we say there is collusion and return the colluding agents, otherwise, we return that collusion could not be detected. Please note that the collusion threshold $\alpha$ is a hyper-parameter that needs to be set using the training and validation data.

# 5 EXPERIMENTS

## 5.1 ROCK PAPER SCISSORS

For the fully observable simultaneous game, we consider a three-player version of iterated Rock Paper Scissors (RPS) where we have all the three players pick an action from either Rock($R$), Paper($P$) or Scissors($S$) at each timestep.

Table 1: Payoff Matrix for Three-Player Rock Paper Scissors.

Player 3 plays Rock($R$)

| | | Player 2 | |
|---|---|---|---|
| | $R$ | $P$ | $S$ |
| $R$ | $(0,0,0)$ | $(0,1,0)$ | $(1,0,1)$ |
| $P$ | $(1,0,0)$ | $(1,1,0)$ | $(1,1,1)$ |
| $S$ | $(0,1,1)$ | $(1,1,1)$ | $(0,0,1)$ |

(Player 1 rows)

Player 3 plays Paper($P$)

| | | Player 2 | |
|---|---|---|---|
| | $R$ | $P$ | $S$ |
| $R$ | $(0,0,1)$ | $(0,1,1)$ | $(1,1,1)$ |
| $P$ | $(1,0,1)$ | $(0,0,0)$ | $(0,1,0)$ |
| $S$ | $(1,1,1)$ | $(1,0,0)$ | $(1,1,0)$ |

(Player 1 rows)

Player 3 plays Scissors($S$)

| | | Player 2 | |
|---|---|---|---|
| | $R$ | $P$ | $S$ |
| $R$ | $(1,1,0)$ | $(1,1,1)$ | $(1,0,0)$ |
| $P$ | $(1,1,1)$ | $(0,0,1)$ | $(0,1,1)$ |
| $S$ | $(0,1,0)$ | $(1,0,1)$ | $(0,0,0)$ |

(Player 1 rows)

As in the common version of the game, we have $R$ beats $S$, $S$ beats $P$, and $P$ beats $R$. The payoffs for the three-player version we consider are decided according to the following rules:

1. If all three actions are the same, each player gets a 0.

2. If all three actions are distinct, each player gets a 1.

3. If there are 2 distinct actions, in the set of actions selected, the winning action is decided according to the general rule stated above. All winners get one point each.

The payoff matrix for the three-player Rock, Paper, Scissors is given in Table 1.

**Manual Collusion:** We implement a simple assistant collusion tactic for the three-player iterated RPS. As described in Section 1, this form of collusion involves a primary colluding agent and a secondary colluding assistant that selects an action that benefits the primary agent. Figure 1 gives a graph representation of assistant collusion for the fully observable simultaneous game. The solid arrows depict the information directly accessible to each agent. $s_t$ denotes the state of the environment at time $t$ and $a_t^i$, $a_t^j$ and $a_t^k$ denote the action taken by agents $i$, $j$ and $k$ at time $t$ respectively. The dashed arrow depicts the covert flow of information between the colluding agents $i$ and $j$, with $i$ being the primary agent and $j$ being the colluding assistant.

For RPS, we can see from Table 1, whenever player 1 chooses $R$ and player 2 chooses $S$, regardless of what action player 3 selects, player 1 always gets a point. In this scenario, player 1 is the primary colluding agent and player 2

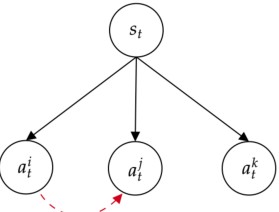

Figure 1: Assistant Collusion for Fully Observable Simultaneous Game.

2 is the colluding assistant. We assume that there is some form of ex-ante coordination or signaling that takes place between the colluding partners to carry out the collusion. We have the following three players for the manual case:

1. **Player A** : Primary colluding agent.
2. **Player B** : Assistant colluding agent.
3. **Player C** : Non-colluding agent.

Player A and Player C choose an action at random. Player B chooses an action that guarantees A a point with some *collusion probability*, and a random action otherwise. In a practical setting, the colluding partners may not collude on every move to avoid suspicion. The *collusion probability* $(CP)$ governs the probability of active collusion, e.g. if $CP = 0.4$, the assistant plays a move that benefits the primary agent for $40\%$ of the games and plays a random move for the other $60\%$ games. In the case of RPS, we note that the higher the $CP$, the higher the win rate for the primary colluding agent.

**Automatic Collusion:** To test our method on other collusion strategies, we train the players to learn to collude automatically. For this, we make use of Deep Reinforcement Learning. Specifically, we treat the two colluding players as a single agent, with a joint state and action space. We provide the state space specification in Appendix A. We utilize the Double Deep Q-Network (DDQN)[Van Hasselt et al., 2016] for training. The agent receives a reward of +1 if one of the colluding players wins and a -1 if the non-colluding player wins. We have the following three players for automatic collusion:

1. **Player D** : Auto-colluding agent.
2. **Player E** : Auto-colluding agent.
3. **Player F** : Non-colluding agent. This agent selects an action uniformly at random from valid actions.

For all the experiments collusion threshold is set at $0.05$.

**Experiment 1:** For the first experiment we attempt to answer the question: How does collusion strength affect the swiftness or sample complexity of our detection algorithm? We use data generated from games played between Players A, B, and C (manual collusion). We run multiple simulations for a different number of games (sample size) and varying levels of collusion probability values. We plot the calculated net influence for different settings in Figure 2. Each graph in the figure is generated for the different $CP$ values. The y-axis gives the net influence values and the x-axis gives the number of games used to calculate the net influence values. The dashed horizontal line in each graph depicts the collusion threshold $\alpha$ which is set at $0.05$. Note that, as the $CP$ values go higher, our algorithm can detect collusion using data from fewer games. However, we also note that we are not able to detect collusion for the case where $CP = 0.1$, and $\alpha = 0.05$ irrespective of the sample size. Please note our method can detect collusion if $\alpha$ is set to a lower value. We provide additional results for a 4 player version of the game in Appendix C

We plot the net influence values calculated from data generated from 1000 games for varying levels of collusion probabilities in Figure 3. The y-axis gives the net influence values for all pairs of players and the x-axis gives the $CP$ values. We note that we can detect collusion in cases where the collusion probability is over $0.2$. From Figure 3, we observe that as the level of collusion between players A and B strengthens, the net influence they have on each other also increases. This indicates that the value of the net influence could possibly indicate the level of collusion.

**Experiment 2:** For the second experiment, we determine the collusion detection accuracy (CDA) of our method for varying levels and different forms of collusion. We generate data for a different number of games for both manual and automatic collusion cases. For the manual case with data generated for games played between players A, B, and C, we vary the $CP$ values as we did for the first experiment. For the automatic case, we generate data from games between players D, E, and F. To get a robust measure of the CDA, we run 1000 iterations per number of games. The CDA gives the percentage of iterations that the algorithm can detect the collusion correctly. We report the results in Table 2. The first column gives the number of game records used to detect collusion. The second and the third column gives the CDA for two of the manual cases ($CP = 0.3$ and $CP = 0.4$) and the fourth column gives the CDA values for the automatic case. For $CP = 1$, we only require data from $60$ games to get a CDA of $100\%$. For $CP = 0.2$, we get a CDA of $24.3\%$ for 1000 games. We generated data from 10000 games to see how that affects the CDA. We get a CDA of $82.3$ for the manual case where $CP = 0.2$. This shows that given enough samples, we can detect collusion even for cases where the strength of collusion is low. However, we only get a CDA of $\approx 8\%$ on average for the case where $CP = 0.1$, again implying that there exists a minimum collusion strength below which our algorithm is unable to detect collusion with high probability.

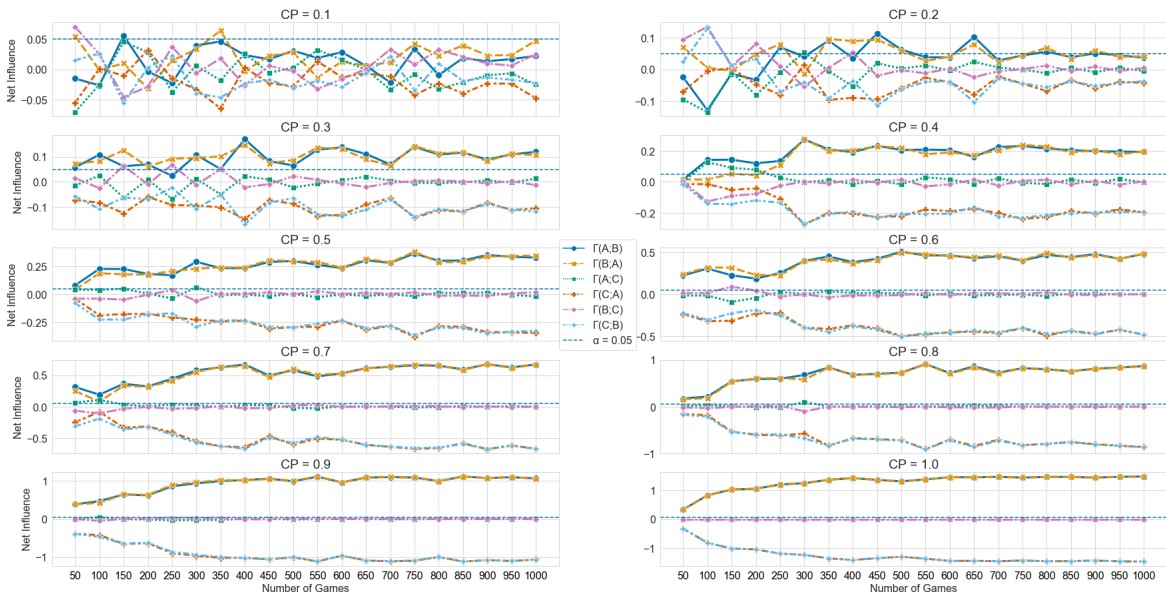

Figure 2: Net Influence Calculated for a Different Number of Games for Varying Values of Collusion Probability ($CP$) for Rock Paper Scissors.

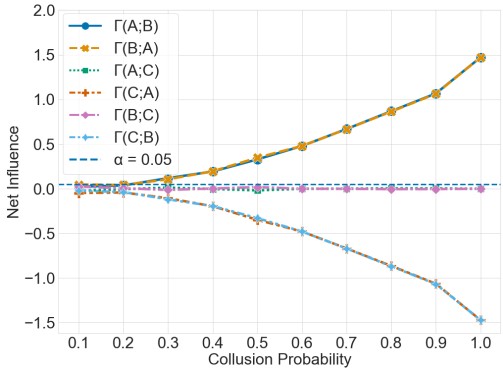

Figure 3: Net influence for Varying Collusion Probabilities ($CP$) over 1000 Games of Rock Paper Scissors.

Table 2: Collusion Detection Accuracy for Rock Paper Scissors across Different Number of Games.

| Number of Games | Manual CDA (%) ($CP = 0.3$) | Manual CDA (%) ($CP = 0.4$) | Automatic CDA (%) |
|---|---|---|---|
| 100 | 37.1 | 58.4 | 81.5 |
| 200 | 48.3 | 79.2 | 93.2 |
| 300 | 71.0 | 96.2 | 98.1 |
| 400 | 82.8 | 99.3 | 98.9 |
| 500 | 89.6 | 99.7 | 99.4 |
| 600 | 94.6 | 99.9 | 99.2 |
| 700 | 97.8 | 100 | 99.2 |
| 800 | 98.8 | 100 | 99.7 |
| 900 | 99.4 | 100 | 99.9 |
| 1000 | 100 | 100 | 99.9 |

## 5.2 LEDUC HOLD'EM POKER

To test our method for partially observable sequential games, we consider a three-player version of Leduc Hold'em poker [Southey et al., 2012]. Leduc Hold'em poker is a simpler variant of poker played with a deck of six cards with three ranks and two suites. For our implementation, we use the ace, king, and queen. At the beginning of the hand, each player antes one chip and is dealt with a private (or hole) card. Following this, there is a round of betting known as the pre-flop betting round. After the first betting round, another card is dealt face-up as a community (or board) card. There is a two-bet maximum per round. The raise size is set at two chips for the pre-flop and four chips for the post-flop betting round. If a player's hole card is the same rank as the board card, they win the pot; otherwise, the player whose private card has the higher ranked card wins the pot. The players are rotated at the end of each hand. Each game goes on for nine rotations, with each player getting to be the dealer thrice. The player positions are shuffled at random before the beginning of each game.

**Manual Collusion:** We develop two colluding agents that follow an association collusion strategy. Recall that an association collusion strategy is one in which both the colluding partners are in a symbiotic relationship and play to each other's advantage. Figure 4 shows a graph representation of association collusion in a partially observable sequential game. $s_t$ denotes the state of the environment at time $t$ and $s_t^i$, $s_t^j$, and $s_t^k$ denote the observed states for each agent $i$, $j$, and $k$ respectively. $a_t^i$, $a_t^j$ and $a_t^k$ denote the actions taken by agents $i$, $j$, $k$ at time $t$ respectively. Note that $t$ depicts one instance of the game, e.g. one round in poker. There is a sequential order ($i \rightarrow j \rightarrow k$) in which the players are

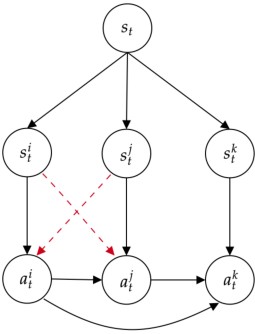

Figure 4: Association Collusion for partially observable sequential game.

Table 3: Net influence Values for Different Levels of Collusion for Leduc Hold'em.

| Players (P1, P2, P3) | Avg. Payoff (P1, P2) | $\Gamma(P1;P2)$ $\Gamma(P2;P1)$ | $\Gamma(P3;P2)$ $\Gamma(P2;P3)$ | $\Gamma(P1;P3)$ $\Gamma(P3;P1)$ |
|---|---|---|---|---|
| A1, A2, A3 | 0.0472 | 0.0032 -0.0033 | 0.0025 -0.0032 | 0.0033 -0.0025 |
| B1, A1, A2 | 1.3874 | -0.0424 0.0026 | 0.0425 0.0424 | -0.0026 -0.0425 |
| B1, B2, A1 | 2.3701 | -0.0761 -0.0740 | 0.0032 0.0761 | 0.0740 -0.0032 |
| B1, B2, B3 | 0.0324 | 0.0001 0.0011 | 0.0011 -0.0001 | -0.0011 -0.0011 |
| C1, C2, A1 | 3.9513 | **0.1133** **0.1168** | 0.0008 -0.1133 | -0.1168 -0.0008 |
| C1, C2, B1 | 1.3302 | **0.1213** **0.1204** | 0.0012 -0.1213 | -0.1204 -0.0012 |
| D1, D2, A1 | 2.3963 | **0.4913** **0.3829** | 0.0487 -0.4913 | -0.3829 -0.0487 |
| D1, D2, B1 | 0.0374 | **0.6808** **0.6797** | 0.0912 -0.6808 | -0.6797 -0.0912 |
| E1, E2, A1 | 8.4340 | **0.2108** **0.3916** | 0.0605 -0.2108 | -0.3916 -0.0605 |
| E1, E2, B1 | 6.5372 | **0.5504** **0.5856** | 0.0314 -0.5504 | -0.5856 -0.0314 |
| D1, D2, F1 | 1.2136 | **0.6527** **0.6150** | -0.0089 -0.6527 | -0.6150 0.0089 |
| E1, E2, F1 | 4.4832 | **0.4317** **0.4606** | -0.0006 -0.4317 | -0.4606 0.0006 |

allowed to make their moves. As stated in Section 3.2, each player has access to the information about the actions of the preceding players. The dashed line depicts information being shared covertly. In Figure 4, we see that agents $i$ and $j$ are the colluding partners. We see that $i$ has access to the observed state of agent $j$ which has a direct effect on their action choice. The same can be observed for $j$. We assume that the colluding partners exchange hidden information covertly or use some form of ex-ante coordination or signaling. For the three-player Leduc Hold'em environment, we develop rule-based agents that have access to the private card of the colluding partner. In the first round, if either of the colluding partners has an ace, both agents raise. In the second round, if either of the colluding partners has a pair or an ace, both agents raise. In all other circumstances, the agents call, if possible. A player only folds if they don't have enough chips to call.

**Automatic Collusion:** For automatic collusion, we develop two collusion strategies - high payoff (HP) and low payoff (LP). For HP, as the name suggests, we train the agents to maximize the payoff they receive at the end of a game of poker. On the other hand, for LP, we train the agents to keep the payoff as close to zero as possible. We use the same method we did for RPS to generate collusion. For training, we treat both colluding partners as a single agent with a joint state and action space. We provide the state space specification in Appendix A. For HP, agents receive a reward of zero during the game and a reward of the sum of the loss or profit each individual player makes at the end of a hand. For LP, the agents receive a reward of zero during the game and a penalty for straying from a net-zero payoff. We train the agents using DDQN for 10000 games played against an agent that chooses their actions at random. The LP collusion strategy emulates players that may be colluding, but the colluding strategy itself may not yield a high payoff. Such collusion would be almost impossible to detect if we were to only look at the payoffs.

To test our method for the case where the non-colluding opponent has a non-trivial policy, we train a learning-based non-colluding agent using DDQN. We design the state-space to include the historical actions of other players. This ensures the policy depends on the observed actions of the other players.

We have the following players for Leduc Hold'em:

1. **Players A1, A2 and A3** : Non-colluding random agents. This agent selects an action uniformly at random from valid actions.

2. **Player B1, B2 and B3** : Non-colluding rule-based agents. This agent raises if they have an ace or a king in the first round and raises only if they have a pair in the second round. For all other cases, the agent selects a random action.

3. **Players C1 and C2** : Associate colluding agents.

4. **Players D1 and D2** : LP auto-colluding agents.

5. **Players E1 and E2** : HP auto-colluding agents.

6. **Player F1** : Learning based non-colluding agent.

For all the experiments collusion threshold is set at $0.05$.

**Experiment 1:** For the first experiment we generate data for 1000 games of Leduc Hold'em poker for different player combinations. The different player combinations we use are given in the first column of Table 3. The first four rows show the results for the case where there is no collusion between players. The last eight rows show results for settings where two of the players are colluding. We report the average payoff per game for two of the players in the second column.

Please note that the orderings of the players as given in the table has no bearing on the results. As stated earlier, we shuffle the player positions before each game and each game consists of 9 rotations. We report the pair-wise net influence of players in Table 3. Recall from Equation (5), when the net influence that two players have on each other exceeds the collusion threshold, we say that the two players are colluding. The values ($\geq 0.05$) for which our method detects collusion are given in bold. We note that our method can successfully detect collusion in all colluding scenarios even in cases where the payoff may be negligible.

**Experiment 2:** We run the second experiment to determine the CDA and the swiftness of our method. Swiftness indicates the number of game records needed to detect collusion with high probability. Recall, CDA gives the percentage of iterations that the algorithm was able to detect the collusion correctly. To check the swiftness, we generate data for a different number of games and run the detection algorithm. To get a robust measure, we run 1000 iterations per number of games. We generate data for association, LP-auto, and HP-auto collusion for two cases: against a random agent (Player A1) and against the rule-based agent (Player B1).

We report the results in Table 4 and Table 5. Since each game of Leduc Hold'em poker consists of multiple rotations, we also report the number of hands in the data used for testing. Note that for the second case (against Player B1), we change the number of rotations per game to three since the CDA was over 95% for 40 games with nine rotations per game for all cases.

From Table 4, we see that we require data from 200 games (or 1800 hands) to get an accuracy of over $95\%$ for association collusion, but only require data from 120 games (or 1080 hands) to achieve the same detection accuracy for both forms of automatically generated collusion. We believe this is because the strength of collusion is stronger in the automatic cases as compared to the hand-crafted rule-based case. We saw in Section 5.1 that the values of net influence could be an indication of the strength of collusion. We see from Table 3 that the manual association collusion setting (Players C1, C2, A1) has the lowest net influence values. We believe this could be why it requires more samples to get a high CDA when compared to the other two cases.

## 6 RELATED WORK

Over the years, collusion detection has been studied extensively across multiple domains. Majority of the literature focuses on collusion detection in auctions and cartel identification in bidding [Hendricks and Porter, 1989, Porter and Zona, 1993, Schurter, 2017, Wachs and Kertész, 2019]. Hendricks and Porter [1989], Porter and Zona [1993] and Schurter [2017] identify collusive bids in auctions. Wachs and Kertész [2019] presents a network-based framework to

Table 4: Collusion Detection Accuracy (CDA) for Leduc Hold'em for Different Number of Games when Played Against Random Non-Colluding Agent (Player A1).

| Number of Games | Number of Hands | Manual CDA (%) | LP-Auto CDA (%) | HP-Auto CDA (%) |
|---|---|---|---|---|
| 20 | 180 | 0.0 | 56.8 | 58.4 |
| 40 | 360 | 0.10 | 67.8 | 78.3 |
| 60 | 540 | 1.53 | 82.8 | 89.6 |
| 80 | 720 | 6.81 | 90.2 | 96.7 |
| 100 | 900 | 19.0 | 97.8 | 97.3 |
| 120 | 1080 | 41.1 | 98.8 | 97.9 |
| 140 | 1260 | 62.8 | 99.3 | 98.3 |
| 160 | 1440 | 78.3 | 99.9 | 98.6 |
| 180 | 1620 | 88.8 | 100 | 99.5 |
| 200 | 1800 | 93.6 | 100 | 99.6 |

Table 5: Collusion Detection Accuracy (CDA) for Leduc Hold'em for Different Number of Games when Played Against Rule-Based Non-Colluding Agent (Player B1).

| Number of Games | Number of Hands | Manual CDA (%) | LP-Auto CDA (%) | HP-Auto CDA (%) |
|---|---|---|---|---|
| 20 | 60 | 54.2 | 95.2 | 94.6 |
| 40 | 120 | 76.4 | 98.6 | 96.3 |
| 60 | 180 | 82.5 | 99.6 | 98.7 |
| 80 | 240 | 89.1 | 100 | 99.6 |
| 100 | 300 | 93.4 | 100 | 100 |
| 120 | 360 | 97.0 | 100 | 100 |
| 140 | 420 | 97.4 | 100 | 100 |
| 160 | 480 | 98.4 | 100 | 100 |
| 180 | 540 | 98.9 | 100 | 100 |
| 200 | 600 | 99.3 | 100 | 100 |

detect potential cartels in bidding markets. Hespanhol and Aswani [2020] formulate the problem of tacit collusion in algorithmic pricing as an inverse variational inequality and design a hypothesis test to detect collusion.

Stock market trading is another domain where collusion detection has been studied extensively [Palshikar and Apte, 2008, Madurawe et al., 2021, Islam et al., 2009, Cao et al., 2016]. Most of these methods utilize clustering techniques to detect collusion. Islam et al. [2009] propose a Markov clustering algorithm, Palshikar and Apte [2008] use multiple graph clustering algorithms and Madurawe et al. [2021] combine anomaly detection with graph clustering to detect collusion sets in trading data. Cao et al. [2016] find collusive cliques using directed graphs and dynamic programming.

There has been some work done in detecting collusion in other online settings such as crowd-sourcing tasks [KhudaBukhsh et al., 2014, Chen et al., 2020] and online rating systems [Allahbakhsh et al., 2013]. KhudaBukhsh et al. [2014] present methods for detecting non-adversarial collusion by analyzing the similarity of workers' answers while [Chen et al., 2020] propose a collusion detection method based on

the statistical test of the consistency of workers' answers across different crowd-sourcing tasks. In Allahbakhsh et al. [2013], the authors propose a collusion detection algorithm for online ratings based on clustering techniques.

Collusion detection in multi-agent games has been previously studied in [Mazrooei et al., 2013, Laasonen and Smed, 2015, Yampolskiy, 2008, Hamilton, 2011, VanderKnyff et al., 2009]. However, most of these approaches focus on providing a solution for a specific type of game. Hamilton [2011] construct interaction graphs between players and apply graph analysis techniques to detect unusual patterns or structures to detect collusion in round-robin iterated prisoner's dilemma tournament. Unlike our multi-agent setup with $n > 2$ agents, they focus on multiple games played between two players. In Mazrooei et al. [2013], authors propose an automatic collusion detection method applicable to only sequential games. They make use a of collusion table that captures the effect of each player's actions on the utility of all players using automatically learned value functions. They do not consider the case where information is shared between the colluding partners. VanderKnyff et al. [2009] and Laasonen and Smed [2015] propose collusion detection specifically for first-person shooter (FPS) games. VanderKnyff et al. [2009] use principal component analysis to detect the colluding players and Laasonen and Smed [2015] use graph clustering algorithms to detect soft-play in shooter games.

## 7 CONCLUSION

In this paper, we propose a novel method to detect collusion in multi-agent games and provide proof of concept experiments to show its effectiveness. Our experiments show that our method can successfully detect collusion with high probability in perfect information simultaneous games and imperfect information sequential games. As the collusion gets stronger, our method requires a lower number of samples to successfully detect collusion. We also see some evidence that the value of the net influence may indicate the strength of collusion. The relationship between net influence and the strength of collusion needs further investigation. A theoretical study of the collusion threshold is warranted and left as future work. Our method focuses on detecting collusion in multi-agent games where exactly two players collude. The approach could extend to the case of more than two colluding players by using the co-information lattice Bell [2003] to calculate the net influence, instead of mutual information. Detailed evaluations will be the subject of future work. In addition, we anticipate that our method can be extended to other types of multi-agent games, including fully observable sequential games such as Monopoly.

## Acknowledgements

This research is supported, in part, by the Defense Advanced Research Projects Agency (DARPA) and the Air Force Research Laboratory (AFRL) under the contract number W911NF2020003. The views and conclusions contained herein are those of the authors and should not be interpreted as necessarily representing the official policies or endorsements, either expressed or implied, of DARPA, AFRL, or the U.S. Government. We thank our team members on this project for all the discussions to develop this paper. Some of the ideas in this paper are based on our learning from the SAIL-ON meetings.

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
