# OpenReview forum: "Information-Theoretic Approach to Detect Collusion in Multi-Agent Games"
_auai.org/UAI/2022/Conference — UAI 2022 Poster_

### Official Review · Reviewer_DVsz · 2022-03-30

**Q2(1) Originality/Novelty:** 3
**Q2(2) Significance/Impact:** 2
**Q2(3) Correctness/Technical Quality:** 1
**Q2(6) Clarity Of Writing:** 3
**Q6 Overall Score:** 6
**Q8 Confidence In Your Score:** 4

**Q1 Summary And Contributions:**

The paper introduces an information-theoretic approach to detect collusion in multi-agent games. More specifically, a Kullback-Leibler divergence between the joint distribution of actions of pairs of players and the respective marginal distributions is used to measure coordination between players. The authors apply their approach to rule-based and optimized collusions in Leduc hold'em poker and in rock-paper-scissors, and show that their approach can successfully detect collusion in these cases.

**Q2 Assessment Of The Paper:**

More detailed information regarding each of these aspects is given below:

**Q2(4) Quality Of Experiments (Optional):**

2: Fair: The experimental evaluation is weak: important baselines are missing, or the results do not adequately support the main claims.

**Q2(5) Reproducibility:**

3: Good: Key resources (e.g., proofs, code, data) are available and key details (e.g., proofs, experimental setup) are sufficiently well-described for competent researchers to confidently reproduce the main results.

**Q3 Main Strengths:**

The approach to collusion detection is intuitive and accessible, and the experimental setup is very clear. The quantitative results are promising. The authors discuss related work in sufficient detail. I generally like the information-theoretic approach and think that this can be very fruitful, if developed further.

**Q4 Main Weakness:**

My main concern is that the proposed collusion measure does not only measure collusion, but any adaptation to other players' actions. For example, in poker, the optimal action of a non-colluding player also depends on the other players' actions and not only on the current state of the game (e.g., I should fold if I have a bad hand and if a different player raises). As another example, if a player simply copies the actions of a different player, this is certainly not collusion, but the "net influence" will still be large. This is not discussed at all in the paper and severely limits its validity. Further, the net influence seems not to generalize easily to multiple colluding players, which is a serious limitation.

**Q5 Detailed Comments To The Authors:**

- In Sec. 3 you argue about independent policies; independent policies do not imply independent actions, because my policy may require to take other players' actions into account. On the other hand, the assumption of independent actions is unrealistic for most games.
- In and before (3), under the independence assumption, $p(a_i,a_j|s_{ij})$ does not factorize as $p(a_i|s_i)p(a_j|s_j)$, but as $p(a_i|s_{ij})p(a_j|s_{ij})$. Also, by definition (3) is symmetric. To allow for unsymmetric $\gamma(i;j)$, one explicitly needs to take the time (or turn index) $t$ into account in the equation.
- In Sec. 3.2, indeed we have that $p(a_i|s_i)\neq p(a_i|s_{ij})$, and this is what seems to be the main difference between full information and partial information games. This deserves to be spelled out explicitly. However, (3) still makes sense in this setting, but with a different denominator. One could argue that under the independence assumption $p(a_i|s_{ij})=p(a_i|a_j,s_{ij})=p(a_i|a_j,s_i)$, which is similar to the arguments in (6)-(7), but explicit for partial information games.
- In the descriptions of Experiment 1 and 2 in Sec. 5.1, the authors write that they "are not able to detect collusion for the case where CP=0.1, irrespective of the sample size". This is a simple consequence of the $\alpha$ setting; for different settings, different CPs will be detectable. This needs to be discussed in the paper.
- It is not clear if the manual collusion strategy in Sec. 5.2 is "good" in the sense that it yields good pay-offs. Rather, I would suggest to train optimal strategies using DDQN, where each player observes its own state AND the actions of the other players to make decisions.

Minor comments:
- Sec. 2: $\mathbf{a}_t=[a_0,\dots]$ -- why does it start with $a_0$, making the vector of length $n+1$?
- In Table 3, why are the first entries in the last two columns exactly negatives of each other? Why is the pay-off for D1,D2,A1 higher than for A1,A2,A3 if the former trains to get a low pay-off?


EDIT: Score updated based on authors' response.

**Q7 Justification For Your Score:**

The proposed measure for collusion can only detect collusion of two players, and further may detect collusion whenever there is a non-trivial influence of one player's actions on another (even though this influence may be simply because both players try to win). Since these settings are not discussed in detail, the reader is left in doubt about the method's limitations. I thus think that more work needs to go into the paper to remove these concerns.

**Q9 Complying With Reviewing Instructions:**

1: Yes.

---

### Official Review · Reviewer_aVDZ · 2022-04-07

**Q2(1) Originality/Novelty:** 3
**Q2(2) Significance/Impact:** 2
**Q2(3) Correctness/Technical Quality:** 3
**Q2(6) Clarity Of Writing:** 3
**Q6 Overall Score:** 6
**Q8 Confidence In Your Score:** 4

**Q1 Summary And Contributions:**

The paper proposes an information-theoretic approach to detect collusion in multi-players games.
The point of view is that of an external arbitrator that can see the whole history of the game and identify if some players are colluding.
The idea is to use mutual information to recognize if actions of two players are more related than a given collusion threshold.


**Q2 Assessment Of The Paper:**

More detailed information regarding each of these aspects is given below:

**Q2(4) Quality Of Experiments (Optional):**

2: Fair: The experimental evaluation is weak: important baselines are missing, or the results do not adequately support the main claims.

**Q2(5) Reproducibility:**

2: Fair: Key resources (e.g., proofs, code, data) are unavailable but key details (e.g., proof sketches, experimental setup) are sufficiently well-described for an expert to confidently reproduce the main results.

**Q3 Main Strengths:**

The paper proposes an information-thoretic model to identify collusion among players in a multi-player game and give some experiments to support the effectiveness of the method in two specific games.

The paper is clearly written.

**Q4 Main Weakness:**

The model seems too simple and relevant only for some classes of games (see more specific comments in the following).
Experiments are partially convincing.



**Q5 Detailed Comments To The Authors:**

I have several questions about the relevance of the proposed approach for general games. I think it can work only for specific classes of games that the authors should accurately characterize. In particular,

1. Authors assume that if the probability of joint actions of two players is different from the product of their marginal probabilities the they are colluding. I think this does not hold in general. For example, in repeated games where players learn from the history of the game about the type of their  adversaries you will tend to identify a high correlation between thee actions of the players even if they are not colluding.

2. Authors are considering only the case of a collusion between two players and it's not clear if the approach still holds in case of larger coalitions.

3. The collusion detection algorithm is based on the computation of a policy matrix that expresses thee relation between the status of the game and the actions of the players, but it's not clear what the status represents. I'd like the authors should clarify on this point.

Experiments are focused on aa very specific setting: two colluding agents and one competitor. It's not clear if results can be extended t more general settings.


**Q7 Justification For Your Score:**

Collusion detection is a well-studied and high-impact problem in several contexts. However, the main contribution of the paper  has no novelty, both on the theoretical and methodological point of view.
I have several objections on the model and on its relevance to general multi-player games. Experiments are focused n a very specific setting and it's not clear if results hold in general.


**Q9 Complying With Reviewing Instructions:**

1: Yes.

---

### Official Review · Reviewer_EGZ4 · 2022-04-09

**Q2(1) Originality/Novelty:** 3
**Q2(2) Significance/Impact:** 3
**Q2(3) Correctness/Technical Quality:** 3
**Q2(6) Clarity Of Writing:** 4
**Q6 Overall Score:** 8
**Q8 Confidence In Your Score:** 4

**Q1 Summary And Contributions:**

The authors develop a methodology for detecting collusion in multiagent games. The main idea is that if two players collude, then the actions they take (in various states) should be nontrivially correlated. The authors use information-theoretic tools to establish the level of this correlation (or, influence of one player on the other) and test if their approach works experimentally. To this end, they use a simple rock-paper-scissors game (extended to three players) and a variant of poker.

**Q2 Assessment Of The Paper:**

More detailed information regarding each of these aspects is given below:

**Q2(4) Quality Of Experiments (Optional):**

3: Good: The experimental evaluation is adequate, and the results convincingly support the main claims.

**Q2(5) Reproducibility:**

3: Good: Key resources (e.g., proofs, code, data) are available and key details (e.g., proofs, experimental setup) are sufficiently well-described for competent researchers to confidently reproduce the main results.

**Q3 Main Strengths:**

The authors solve a very reasonable problem in a general and compelling way.
The experiments are convincing.

**Q4 Main Weakness:**

I was not sure why in the poker experiments the authors did not use deep learning to train an agent against which the colluding ones would play.

**Q5 Detailed Comments To The Authors:**

Please explain why not train a non-colludiing agent for the poker experiment using deep learning.

**Q7 Justification For Your Score:**

Important problem, good solution, convincing experiments, high quality of writing

**Q9 Complying With Reviewing Instructions:**

1: Yes.

---

### Decision · Program_Chairs · 2022-05-15

**Decision:**

Accept (Poster)

**Comment:**

Meta Review: The paper introduces an information-theoretic measure to detect collusion (=cooperation between subsets of players) in multi-agent games. The measure is based on comparing distributions of joint actions of pairs of players against the marginal distribution over actions (which is motivated from a mutual information viewpoint and implies using the KL-divergence to compare distributions).

Pro:
* A challenging theoretical problem tying into a line of research with significant downstream real-world applications (e.g. collusion in financial markets;)
* A theoretically well motivated approach (taking into account the clarifications during the rebuttal)
* Reviewers agree that the paper is well written
* Compelling experimental results

Cons:
* By far the main criticism (raised by two reviewers independently) before the rebuttal was that the measure is flawed, and could potentially mistake coordination between two players as collusion when instead their actions are correlated simply because they are in an adversarial relationship. The authors' response has clarified this as a misunderstanding.

After the clarification by the authors, aVDZ and DVsz have raised their score to a weak accept. To me personally, the main weakness of the paper has been successfully addressed by the authors and I am in favor of acceptance. There are still some open minor issues, and the paper is probably only highly relevant for a sub-community of UAI - that's why I currently do not see the paper as a candidate for an oral.